# Diagnostic Methods in Forensic Pathology: A New Sign in Death from Hanging

**DOI:** 10.3390/diagnostics13030510

**Published:** 2023-01-30

**Authors:** Maricla Marrone, Gerardo Cazzato, Pierluigi Caricato, Carlo Angeletti, Giuseppe Ingravallo, Nadia Casatta, Carmelo Lupo, Francesco Vinci, Gisella Agazzino, Alessandra Stellacci, Antonio Oliva

**Affiliations:** 1Section of Legal Medicine, Department of Interdisciplinary Medicine, University of Bari “Aldo Moro”, 70124 Bari, Italy; 2Section of Pathology, Department of Precision and Regenerative Medicine and Ionian Area (DiMePRe-J), University of Bari Aldo Moro, 70124 Bari, Italy; 3Innovation Department, Diapath S.p.A., Via Savoldini n.71, 24057 Martinengo, Italy; 4Section of Legal Medicine, Public Health Institute, Catholic University of the Sacred Heart, 20123 Milano, Italy

**Keywords:** sternocleidomastoid muscle, hanging, forensic pathology

## Abstract

Purpose: To evaluate the usefulness of studying vital injuries at the sternal head insertion of the sternocleidomastoid muscle in the medico-legal assessment of death by hanging. Materials and Methods: Study material was obtained from eight bodies of people who died from hanging. The control group included as many specimens collected from people who died from traumatic causes other than hanging (precipitation from medium to large heights and traffic accidents). The structures under study were examined histologically with a BX-51 light microscope (Olympus). An analysis of the extravasated erythrocytes was performed by counting the number per mm^2^ in the histologic section on 10 HPF (400×), and Student’s *t*-test for a comparison of the averages was applied for all parametric values. The authors noted that the key finding, indicative of the subject’s viability at the time of discontinuation, was the presence of recent hemorrhagic infiltrate (in the absence of hemosiderin) at the tendon insertion of the sternocleidomastoid muscle and the proximal part of the muscle itself. Results: All specimens tested were positive for the presence of hemorrhagic infiltrate at the portions tested in a statistically significant manner. In contrast, in the control cases there was no or, where present, no statistically significant (*p* < 0.05) presence of recent hemorrhagic infiltrate. The limitation of the study is the low number of samples examined. In any case, the results obtained are strongly indicative of the possibility of using this type of forensic pathological investigation in cases where there is a doubt in terms of a differential diagnosis between hanging (suicidal type) and suspension of a corpse in a simulation of hanging.

## 1. Introduction

As is widely known, hanging is a form of overinsufflation in which the neck is subjected, through the use of a noose attached to one end, to a force generated by the weight of the body and the force of gravity [1]. Most deaths by hanging are suicidal in nature [2]; accidental hanging or homicide by hanging is less common [3]. In numerous cases of a corpse found hanged, the medical-forensic scientist is called upon to make a differential diagnosis between suicide, homicide, or suspension of the corpse [4].

One of the typical signs of hanging is the presence, usually above the larynx, of a skin furrow caused by tying the noose around the cervical region. Depending on the characteristics of the medium used as a noose, two types of furrow can be distinguished: a soft furrow produced by soft tools that has no excoriations, hemorrhagic ridges, or vesicles and, therefore, will have the characteristics of a faint, light-red depression; the hard furrow, determined by media of greater consistency, is excoriated, parchment-like, and orange or yellowish in color [5,6].

The differential diagnosis between hanging and cadaver suspension involves looking for peculiar features that are widely known to the forensic pathologist:(1)Excoriations and septal hemorrhages at the level of the sulcus, with hemorrhagic extravasations into the dermis, subcutaneous, and interstitial tissues of the neck;(2)Lesions of the carotid intima (Amussat’s sign);(3)Minute hemorrhages of the adventitia of the common carotid artery (Friedberg’s sign);(4)Fragmentation of the myelinated fibers of the vagus nerve (the sign of Dotto);(5)Hemorrhages in the cervical lymph nodes (Jankovich and Incze’s sign);(6)Retropharyngeal and paravertebral ecchymosis produced by compression of the base of the tongue against the pharynx (Brouardel’s sign);(7)The presence of subconjunctival and endocardial hemorrhagic petechiae, indicative of overinsufflation;(8)Hemorrhages below the anterior longitudinal ligament in the dorso-lumbar segment;(9)Tumidity of the penis and spermatorrhea [7].

Other findings such as protrusion of the tongue, fracture of the hyoid bone, lesions of the vertebrae, and “glove” and “trouser” hypostasis arrangements are not invalidating signs [8,9,10].

Injuries to the internal structures of the neck can be caused in two ways: directly, at the point of maximum compression of the ligature, which occurs on the side opposite the knot point; or indirectly by elongation of the neck’s structures, which is most detectable in the area surrounding the noose knot. In a study conducted from January 1997 to January 2002 at the Institute for Forensic Medicine in Belgrade, 175 hanging autopsies were performed. The results show that there is no clear correlation between the injuries and the type of hanging. In particular, spinal and blood vessel injuries, although rare, vary according to the position of the noose: the former are evident in cases of hangings with anteriorly placed nooses, the latter in cases of hangings with posteriorly placed nooses; in both cases, they could be caused by excessive stretching of the neck’s structures.

An emergent and notable fact is that the most frequently found injuries are muscle hemorrhages caused by direct pressure and stretching of these structures [11].

This was not the only study highlighting the importance of hemorrhages of the neck muscle in subjects who died by hanging.

In fact, in another prospective study conducted by Sharma et al., the various injuries of cervical structures in deaths from neck constriction were examined. Of the 1746 medicolegal autopsies conducted during the study period, 5% were deaths due to overinsufflation. Among them, hanging was the most common cause of death (69%) compared with other overinsufflation deaths: ligature and/or manual strangulation, and suffocation. Injury to the sternocleidomastoid muscle (54%) was the most common injury to the structures of the neck.

In 80% of cases, a single loop was observed around the neck; in 58% of cases, this was located above the thyroid gland [12].

While the neck muscles and injuries to them, precisely because they are a frequent finding during an autopsy, can play a central role in the evaluation of cases of hanging, i.e., they can be of assistance to the medical examiner in defining the mode (suicide/corpse suspension), a targeted microscopic histological investigation is important [13].

To date, several studies, through histologic examinations (a valid tool for verifying the viability of hanging and other forms of overinsufflation), have assessed “viable” changes in the muscle fibers, for example, segmental or discoidal fragmentation of the muscle fibers with a loss of sarcoplasmic cross-striation [14].

Numerous studies have also been conducted over the years that have been useful in distinguishing the viability of lesions found on cadavers of subjects found hanged through other modalities (blood component analysis, tissue viability, lung assessment, etc.), as well as the viability of the subject him/herself at the time of the overinsufflation event [15].

However, another avenue of investigation has remained almost unexplored: if the discovery of the corpse does not occur in the immediate period after the death, or if it occurs so long after the death that the transformative putrefactive processes have taken over, what can be useful to analyze to distinguish a death by hanging from a death from some other cause? Are the findings usually analyzed still useful?

These are the questions the authors wished to propose answers to in this study, the idea for which arose precisely as a result of the difficulties encountered in a dubious court case.

## 2. Materials and Methods

### Case Report

The prelude to the study conducted by the authors was a case concerning the discovery of a suspended corpse in an advanced putrefactive state in which the circumstantial datum was doubtful. In particular, due to the extensive postmortem transformative phenomena, the dislocation of the hypostases could not be detected, and the study of the furrow was equally difficult; similarly, the detection of other findings typical of hanging, such as the Amussat sign, was not possible (Figure 1).

The circumstantial historical record, though scanty, described a solitary subject, whose discovery was an unexpected event during a hunting trip.

The autopsy examination allowed the exclusion of traumatic-type injuries (albeit with the limitations arising from the putrefactive state); among the few tissues still present and adhering to the bones was the sternal insertion of the sternocleidomastoid muscle. It was decided to harvest both to evaluate them microscopically.

A histologic examination following the autopsy showed the presence of a hemorrhagic infiltrate at the sternal insertion of both sternocleidomastoid muscles, indicative of the mechanism of traction acting at that site following suspension of the living subject.

Indeed, recall that, generally, the lesions of the blood vessels of the neck (intimal transverse lesions and perivascular hematomas) in hanging, although not always present, allow us to state that there is a greater tendency of ipsilateral blood vessel lesions related to the position of the ligature knot. This finding, therefore, allows us to state that blood vessel injuries, in cases of hanging, are caused by traction, not by direct pressure on the vessel itself [16]. Traction injury correlates well with vital hanging injury. Such hemorrhagic infiltrate, moreover, was also found intermingled with the muscle fibers and in association with discontinuity of the muscle fibers.

On the basis of these few elements, it was decided to close the case as death by suicide hanging, as there was no evidence to suggest any other mode.

This case and the difficulties encountered, however, prompted the authors to evaluate more thoroughly the only sign that could be characteristic: the hemorrhagic infiltrate at the sternal insertion of both sternocleidomastoid muscles.

The aim was to find it in cases of ascertained death by complete hanging and not to find it, on the other hand, in other types of death from other traumatic causes in order to be able to recognize it as a sign indicative of this type of overinsufflation death and also for further confirmation in cases similar to the one already addressed.

Obviously, such a finding had to have the characteristics of recentness and thus had to show acute tissue reactions in order to be distinguished from older traumatisms. In addition, as previously stated, the hemorrhagic infiltrate at that region would have permitted the assertion that there had been traction—and thus suspension—exerted at the anterior regions of the neck.

For this reason, the authors noted that the key finding, indicative of the subject’s viability at the time of hanging, was the presence of recent hemorrhagic infiltrate (in the absence of hemosiderin) at the tendon insertion of the sternocleidomastoid muscle and the proximal part of the muscle itself.

To confirm this finding as indicative of death by plausible hanging, the authors then decided to repeat the same histological investigation on an additional 7 cadavers with a definite diagnosis of hanging, using deaths of a different traumatic nature from hanging as case control.

The positive group thus included 8 corpses (the case mentioned above plus the additional 7 corpses) belonging to subjects who died by hanging that came to the attention of the Institute of Forensic Medicine in Bari and Rome (Cattolica) in 2021. The comparative control group of subjects who died from traumatic deaths other than hanging consisted of the same number of corpses from the same institute.

Specifically, the 8 “positive for hanging” cases all involved young subjects (aged 20 to 50 years), whose death had occurred by complete and typical hanging, and the ligature had a single loop (the groove was, therefore, unique). The controls had the same characteristics as the cases in terms of age group.

For both groups, samples of the sternal head insertion of the sternocleidomastoid muscle were taken at autopsy.

Sections obtained from the groups were fixed in neutral 10% buffered formaldehyde, and properly processed and immersed in kerosene. Sections with a thickness of 5 microns were then obtained by microtome cutting, and routine staining with hematoxylin/eosin was performed. After that, they were analyzed by bright-field light observation with an Olympus BX-51 light microscope supplied to the Institute of the University of Pathological Anatomy, Bari Polyclinic. Special attention was paid to the presence of recent hemorrhagic extravasation; therefore, a semiquantitative anatomo-pathological evaluation was used by counting the number of cells/mm^2^ over 10 fields at high magnification (400×). Since these were parametric-type values, after deriving the averages for both groups, Student’s *t*-test was used to compare averages. A *p*-value less than 0.05 (*p* < 0.05) was considered statistically significant. All statistical analyses were performed using the program Prism 9.2.0, GraphPad Software, La Jolla (San Diego, CA, USA).

In all eight cases of death by hanging, abundant hemorrhagic infiltrate was consistently observed among the muscle fibers of the sternocleidomastoid muscles near the areas of sternal insertion (Figure 2, Figure 3, Figure 4, Figure 5 and Figure 6).

This histologic finding was, on the other hand, completely absent in the control cases, although hemorrhagic infiltrate was sometimes present in the muscle co-test (indicative of direct and not traction trauma).

## 3. Results

The presence of extravasated erythrocytes among the muscle fibers was 42.50 ± 1.48 (cells/mm^2^) in the study patient group and 3.45 ± 1.45 (cells/mm^2^) in the control patient group. The difference between the two groups was statistically significant (*p*-value: <0.0001) (Figure 2).

## 4. Discussion

The medico-legal investigation aimed to answer the classic questions (the cause and means of death and the mode of the event, i.e., homicide, suicide, or accident), which, in cases of death by mechanical overinsufflation, is not always easy [17]. In fact, it required the integration of historical data, circumstantial data, an inspection (external cadaveric examination), and anatomo-pathological data. Identifying the morphological findings of viability is critical in determining whether an injury was inflicted during life or postmortem. Differential diagnosis among hanging, strangulation, and strangulation is still a challenge for the medical examiner, especially when unusual findings are found [18].

Currently, as highlighted in the relevant literature, researchers have not discovered many specific molecular and immunohistochemical markers or valid uses for them with the easier diagnosis of viability in the ligature marks. The only criteria that remain firmly established are microscopic findings that must be framed in a broader context through the experienced eyes of a pathologist [19].

Sometimes, it is only through the integration of medico-legal and circumstantial data that an assessment can be reached, which, however, if lacking, a strong scientific basis may leave interpretive doubts.

The support of an objective pathological element in the corpse supports the outcome of a judicial investigation with greater force and strengthens the conviction on the mode of death. Circumstantial data, though carefully evaluated, can be misleading because they may have been tampered with by the offender her/himself; the support of inferable elements from the cadaver may sometimes be the only resource to resort to [20].

The cadaver can speak even if it is now dead; one just needs to understand its “language”, which is why the authors decided to consider the possibility that it was the tissues most “resistant” to the transformative and destructive postmortem mechanisms that could provide the answers to investigative doubts [21].

Most of the findings in the corpses of hanged persons, in fact, do not depend on the specific cause of death but, more generically, on violent death or even just suspension, regardless of whether this was of a corpse or a living person: even in the former case, in fact, the most obvious features such as the sulcus, protrusion of the tongue, turgidity of the penis, and hypostases in typical locations are found [18].

It is evident that the finding of hypostases in typical locations (gloves and trousers) and, at the same time, in other locations, will have to raise the suspicion that the corpse was “hanged” after death, which occurred with a sufficiently long antecedent time, to determine the persistence of hypostasis in the primary locations [22].

It is equally evident that the finding of “classic” signs such as Amussat’s, can easily direct the forensic physician toward a death by hanging, but what if all these elements are missing? Can a death remain unresolved or be resolved with irremediable doubts?

The study proposed by the authors originates precisely from a case in which it was necessary, in the face of the controversial historical data, to establish the cause of death (overinsufflation or other) and whether suspension of the corpse could be suspected.

The advanced putrefactive stage of the corpse made the study of the classic signs of death by overinsufflation in general and hanging in particular complex [23]. Even more complex, therefore, was the differential diagnosis between hanging and corpse suspension.

Because of the extensive postmortem transformative phenomena, dislocation of the hypostases could not be detected, and the study of the sulcus was equally difficult; hemorrhagic ridges or serous vesicles were not detectable. Petechiae at the bulbar or tarsal conjunctiva were also not present [24].

A careful evaluation of all the available parameters was therefore necessary. In this regard, since it was not possible to harvest the sternocleidomastoid muscle near the sulcus, the authors proceeded to harvest the sternal insertion of the same (the only portion that was suitable for study). The histologic investigation showed a hemorrhagic infiltrate (intermingled with the muscle fibers and in association with the discontinuity of the same) at this insertion, indicative of the mechanism of traction compatible with suspension of a living subject [25].

A comparison with other corpses of subjects who had died from suicidal hanging, as well as with a control group (homogeneous in terms of age group) allowed the confirmation of the persistence of the identified finding. The hemorrhagic infiltrate at the level of the insertion of the sternocleidomastoid bilaterally was a constant finding in deaths from hanging and was completely absent in other traumatic injuries.

When a corpse therefore can no longer show most of the pathognomonic signs of hanging, it can still speak to us through the sternocleidomastoid insertion. In other words, even when putrefaction is now advanced and has destroyed much of the tissue that is useful for assessments known in the literature, it is possible to evaluate a new finding [19].

Certainly, a limitation of the proposed study is the limited number of samples, which the authors propose to expand in later analyses.

The involvement of multiple national and possibly international reference centers may provide strong statistical data and possibly support this new finding.

## 5. Conclusions

In all cases where the medical examiner is confronted with the need to make a differential diagnosis between death by hanging and suspension of a corpse, such a diagnosis may not be easy on the basis of common autopsy and on-the-spot investigations in association with the historical-circumstantial data [26]. Putrefactive transformative phenomena, and the possible action of environmental flora and fauna could eliminate important biological data, in the absence of which, the forensic scientist should be able to succeed in making a diagnosis using other resources [3,17,27,28].

The study proposed by the authors thus suggests the usefulness of the histopathological study of the sternal insertion of the sternocleidomastoid muscle and thus adds another diagnostic possibility to those already classically recognized and used in such circumstances.

This additional sign is, therefore, of fundamental use to the medical examiner, along with the overall evaluation of the available technical elements, to place or exclude suspicion of death of another nature and/or suspension of a corpse in doubtful cases.

## Figures and Tables

**Figure 1 diagnostics-13-00510-f001:**
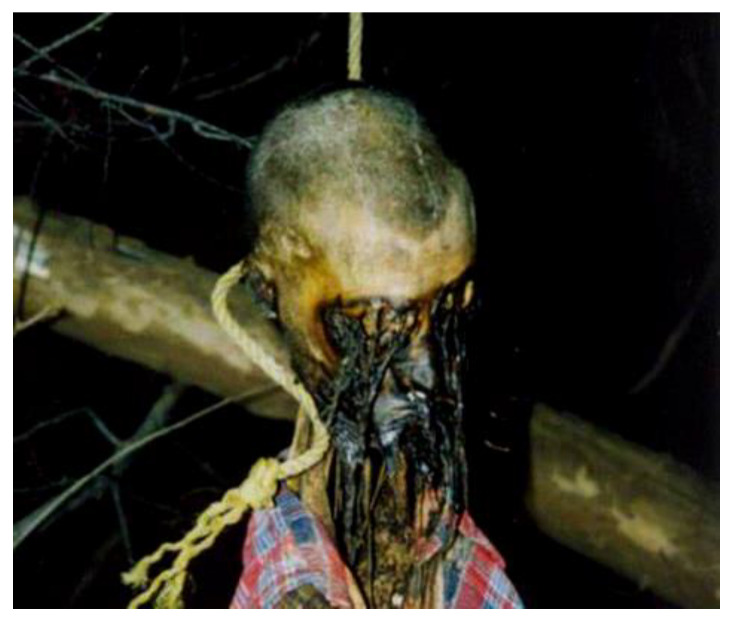
Corpse found with extensive putrefactive phenomena that did not allow the evaluation of common parameters for formulating a diagnosis of hanging.

**Figure 2 diagnostics-13-00510-f002:**
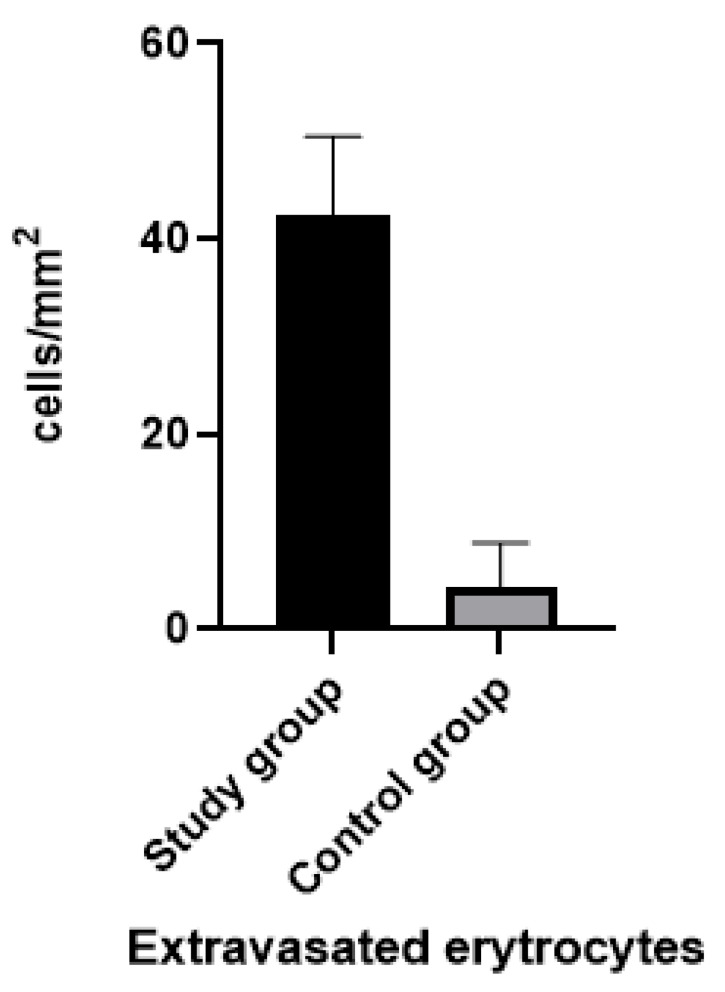
Graph showing the difference between the two study groups. Note that there was a statistically significant difference (*p* < 0.001).

**Figure 3 diagnostics-13-00510-f003:**
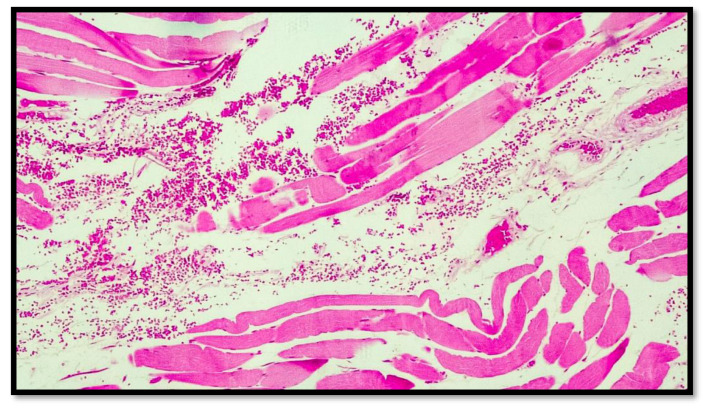
Histological preparation showing the presence of hemorrhagic infiltrate between the skeletal muscle fibers near the sternal insertion head of the sternocleidomastoid muscle. Note the presence of misalignment between the muscle fibers due to the mechanical traction force (hematoxylin–Eosin, Diapath S.p.A.). Original magnification, 4×.

**Figure 4 diagnostics-13-00510-f004:**
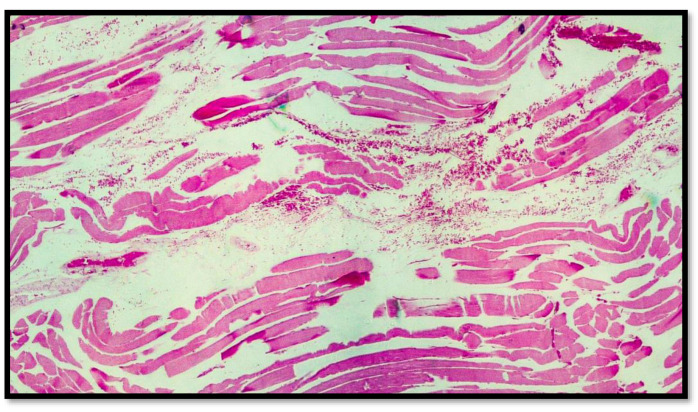
Histological preparation belonging to another subject showing the same characteristics mentioned in Figure 2. Note the presence of the blood red cells interspersed between the muscle fiber cells with the presence of segmentation of the muscle compartment (hematoxylin–eosin, Diapath S.p.A.). Original magnification, 4×.

**Figure 5 diagnostics-13-00510-f005:**
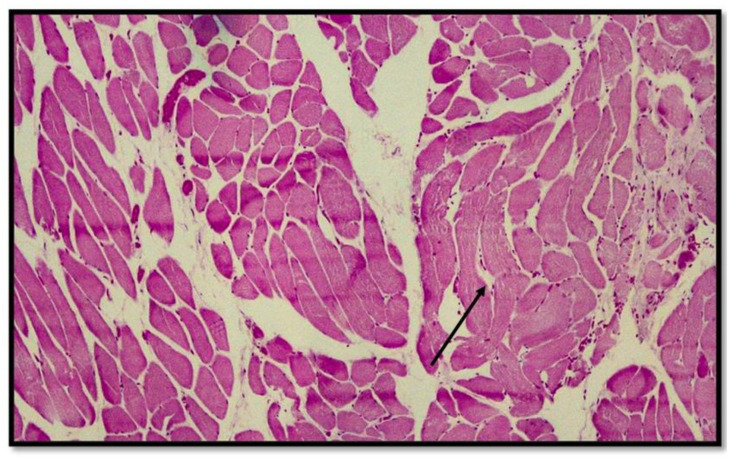
Photomicrograph showing more detailed striated muscle fiber cells with interspersed red blood cell extravasations (black arrow) (hematoxylin–Eosin, Diapath S.p.A.). Original magnification, 10×.

**Figure 6 diagnostics-13-00510-f006:**
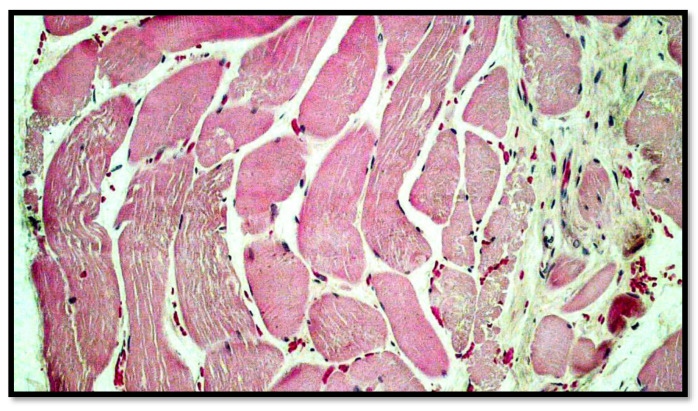
Routine preparation showing, at higher magnification, the presence of extravasated red blood cells between the striated muscle fibers, and the presence of connective tissue between the muscle fibers (hematoxylin–Eosin, Diapath S.p.A.). Original magnification, 20×.

## Data Availability

Not applicable.

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
