# Peer review of "Diagnostic Methods in Forensic Pathology: A New Sign in Death from Hanging"

_diagnostics, 2023, doi:10.3390/diagnostics13030510_

Round 1

Reviewer 1 Report

I enjoyed reading your manuscript and found it informative, interesting and useful. I made a few suggested minor edits and found no significant errors in content, findings or conclusions. I think the information provided in this manuscript will be of importance to the medicolegal community.   

Author Response

Dear Reviewer, thank you very much for your wonderful words. We have corrected all. Thank you.

Reviewer 2 Report

1.      Only the  hemorrhage in form of excoriations are mentioned. Why weren’t septal hemorrhages considered as well?

2.      My advice is to use the term “overinsufflation” when discussing asphyxia.

3.      All the terms used should be standardized  and put in line with the literature ; may I suggest an excellent  reference work – Shkrum, Michael J., and David A. Ramsay. Forensic pathology of trauma. Springer Science & Business Media, 2007., no bias

4.      Histology slides/ images are poorly described – in particular – at fig. 5 – “…preparation showing, at higher…” I do not understand this sentence.

5.      Starting a caption with “another” seems pretty recess.

6.      Please, increase the number of references

7.      Avoid vague and informal expressions 

Author Response

Dear Reviewer n'2,

thank you very much for your kind and useful tips. We have corrected all in order to your suggestions. We hope that it will be fine.

Thank you very much

Round 2

Reviewer 2 Report

I have zo admit that this version is much more fluent. Number of references is optimal. Though, in captions of figs 3-5, bothing has been changed. As I know how bitter I am when I recieve a nonspecific and vague reviwer's note, I+ll try to be as specific as possible - e.g., ".... Routine preparation showing, at higher magnification, the presence of extravasated red blood 451 cells between striated muscle fibers..." - what organ? How do you usually describe an HE slide? Ask a pathoogist for help.

Author Response

I have zo admit that this version is much more fluent. Number of references is optimal. Though, in captions of figs 3-5, bothing has been changed. As I know how bitter I am when I recieve a nonspecific and vague reviwer's note, I+ll try to be as specific as possible - e.g., ".... Routine preparation showing, at higher magnification, the presence of extravasated red blood 451 cells between striated muscle fibers..." - what organ? How do you usually describe an HE slide? Ask a pathoogist for help.

Answer n'1: Dear Reviewer n'2, thank you very much for this useful kinds. We have added some histological informations about the features of this samples. We hope it will be fine now.